# Validity and Reliability of Questionnaires That Assess Barriers and Facilitators of Sedentary Behavior in the Pediatric Population: A Systematic Review

**DOI:** 10.3390/ijerph192416834

**Published:** 2022-12-15

**Authors:** Guilherme Augusto Oliveira, Andressa Costa Marcelino, Maíra Tristão Parra, Marcus Vinicius Nascimento-Ferreira, Augusto César Ferreira De Moraes

**Affiliations:** 1Department of Epidemiology, School of Public Health, University of Sao Paulo, Sao Paulo 01246-904, Brazil; 2Undergraduate Course in Nutrition, Ninth of July University, Sao Paulo 01504-001, Brazil; 3Hebert Wertheim School of Public Health and Human Longevity Science, University of California, San Diego, CA 92093, USA; 4Health, Physical Activity and Behavior Research (HEALTHY-BRA) Group, Universidade Federal do Tocantins, Palmas 77650-000, Brazil; 5The University of Texas Health Science Center at Houston, School of Public Health Austin Campus, Department of Epidemiology, Human Genetics, and Environmental Science, Michael & Susan Dell Center for Healthy Living, Austin, TX 78701, USA

**Keywords:** validity, reliability, questionnaires, barriers, facilitators, sedentary behavior

## Abstract

We systematically reviewed the literature about the validity and reliability of barriers and facilitators of sedentary behavior questionnaires for children and adolescents, considering accelerometers as the reference method. We included studies that assessed the agreement between the barriers and facilitators of sedentary behavior through a questionnaire and an objective measure (e.g., accelerometry). We searched four electronic databases (MEDLINE/PubMed, CINAHL, Web of Science, and SCOPUS): these databases were searched for records from inception to 5 March 2021, and updated to November 2022. The search strategy used the following descriptors: children and adolescents; barriers or facilitators; questionnaires; accelerometers; and validation or reliability coefficient. Studies identified in the search were selected independently by two reviewers. The inclusion criteria were: (i) population of children and adolescents, (ii) original studies, (iii) subjective and objective measurement methods, (iv) studies that report validity or reliability, and (v) population without specific diseases. Seven studies were eligible for our review. The main exclusion reasons were studies that did not report validity or reliability coefficients (56.6%) and non-original studies (14.5%). The participants’ ages in the primary studies ranged from 2 to 18 years. Cronbach’s alpha coefficient was the most reported reliability assessment among the eligible articles, while Pearson and Spearman’s coefficients were prevalent for validity. The reliability of self-report questionnaires for assessing sedentary behavior ranged from r = 0.3 to 1.0. The validity of the accelerometers ranged from r = −0.1 to 0.9. Family environment was the main factor associated with sedentary behavior. Our findings suggest that questionnaires assessing the barriers and facilitators of sedentary behavior are weak to moderate. PROSPERO Registration (CRD42021233945).

## 1. Introduction

Sedentary behavior (SB) is defined as a waking state in which one is sitting or reclining with low energy expenditure (<1.5 METs) [1], for a prolonged time. It has been shown to be one of the main aggravating factors for health issues. Barriers are defined as obstacles that prevent, for the most part, SB, such as the encouragement of physical activity, and limited exposure time to screens. Facilitators are defined as practices that contribute to SB, such as the lack of encouragement to perform outdoor activities, and allowing prolonged screen exposure time [2,3,4].

Sedentary behavior is linked to technological advances through the prolonged exposure to television screens, video games, computers, and smartphones among the young population, and it has significantly increased in recent years. Additionally, sedentary behavior is strongly associated with obesity, increased blood pressure, and cholesterol in the pediatric population [5]. Since the 1980s, obesity has increased in more than 70 countries: more than 107.7 million children were overweight in 2015 [6]. In 2016, the obese pediatric population exceeded 124 million individuals [7]. Besides sedentary behavior being associated with obesity, sedentary behavior is also a risk factor for cardiovascular diseases and early mortality in adulthood [4,8,9].

A previous systematic review on this topic identified that the barriers and facilitators of SB are directly associated with demographic, biological, environmental, and psychosocial factors, including prolonged sitting time, especially when exposed to screens and other media. Neighborhood safety, climate, types of transportation (active, such as walking, or passive, such as transportation by cars) are environmental factors, while social support, that is, people’s influence on social life, is a psychosocial factor associated with SB [3].

The literature regarding questionnaires to assess SB in children and adolescents appears to be scarce. Such instruments allow for large-scale use, and they are more accessible compared to other assessment methods, such as accelerometry. Additionally, questionnaires capture not only SB, but also detailed information regarding SB (e.g., such as mode of SB) and physical activity (e.g., physical activity domain), which are not captured through objective accelerometry data [10]. However, the questionnaires available in the literature are not considered reliable and valid. Therefore, a deeper evaluation of these instruments is warranted for reliable results [11,12,13]. The objective of our review was to assess the validity and reliability of questionnaires addressing the perceived barriers and facilitators of SB in children and adolescents aged 2 to 19 years. Moreover, our review intended to identify the main factors associated with SB. 

## 2. Methods

We conducted a systematic review of the literature about the validity and reliability of questionnaires to assess the barriers and facilitators of perceived SB in children and adolescents. We followed the reporting guidelines of the Preferred Reporting Items for Systematic Reviews and Meta-Analyses (PRISMA). The review protocol is available at PROSPERO (CRD42021233945), and the PECO (population, exposure, comparator, and outcome) research question structure is described below: 

P = children and adolescents

E = barrier & facilitator measured by subjective methods

C = barrier & facilitator measured by objective methods

O = agreement & disagreement reported by statistical methods

### 2.1. Eligibility Criteria 

Eligible studies were those meeting the following inclusion criteria: (i) the study was a primary research study; (ii) the study population consisted of children (2–10 years) and adolescents (11–19 years), as defined by the World Health Organization (WHO) [14]; (iii) the study applied a subjective and objective method to assess SB; (iv) of the study reported the validity or reliability of questionnaires; (v) the study was conducted on healthy populations (free of known diseases). Publications related to the same study were pooled and we considered the publication with the largest sample size and the first publication date. 

### 2.2. Search Strategy

Four electronic databases—MEDLINE (via PubMed), CINAHL, Web of Science, and SCOPUS—were searched from inception to 5 March 2021. The searches were registered in the National Center for Biotechnology Information (US National Library of Medicine, Bethesda, MD, USA) so that continual updates on new publications would be received until 11 November 2022 as requested by the reviewer. We used the descriptors and MeSH terms described below, where List A refers to studies with children, and List B to studies with adolescents.

#### 2.2.1. List A

(‘early childhood’ OR ‘child’ OR ‘preschool’ OR ‘children’ OR ‘preschoolers’ OR ‘childhood’) AND (‘Sedentary Behaviors’ OR ‘Sedentary Lifestyle’ OR ‘Inactivity Physical’ OR Sedentary Time’ OR ‘Lack of Physical Activity’ OR ‘screen time’ OR ‘television’ OR ‘computers’ OR ‘video games’) AND (‘Barrier’ OR ‘physical barriers’ OR ‘Barrier, Physical’ OR ‘Barriers Physical’ OR ‘limitations’ OR ‘facilitator’) AND (‘questionnaire’ OR ‘self-report OR ‘proxy report’ OR ‘log’) AND (‘accelerometer’ OR ‘accelerometry’ OR ‘direct observation’ OR ‘pedometer’ OR ‘motion sense’ OR ‘heart rate’ OR ‘inclinometer’ OR ‘activity monitor’ OR ‘ActiGraph’ OR ‘GENEActiv’) AND (‘validity of results’ OR ‘validities’ OR ‘valid’ OR ‘validation’ OR ‘validity’ OR ‘agreement’).

#### 2.2.2. List B

(‘adolescence’ OR ‘adolescents’ OR ‘youth’ OR ‘teen’ OR ‘teenager’) AND (‘Sedentary Behaviors’ OR ‘Sedentary Lifestyle’ OR ‘Inactivity Physical’ OR Sedentary Time’, OR ‘Lack of Physical Activity’ OR ‘screen time’, OR ‘television’, OR ‘computers’ OR ‘video games’) AND (‘Barrier’ OR ‘physical barriers’ OR ‘Barrier Physical’ OR ‘Barriers, Physical’ O ‘limitations’ OR ‘facilitator’) AND (‘questionnaire’ OR ‘self-report’ OR ‘proxy report’ OR ‘log’) AND (‘accelerometer’ OR ‘accelerometry’ OR ‘direct observation’ OR ‘pedometer’ OR ‘motion sense’ OR ‘heart rate’ OR ‘inclinometer’ OR ‘activity monitor’ OR ‘ActiGraph’ OR ‘GENEActiv’) AND (‘validity of results’ OR ‘validities’ OR ‘valid’ OR ‘validation’ OR ‘validity’ OR ‘agreement’).

### 2.3. Data Extraction

We removed duplications, and two reviewers (GO and AM) independently screened the titles and abstracts. In the second phase, the same reviewers independently screened the studies in full text using the eligibility criteria presented in Figure 1. Differences were resolved through discussion until a consensus was reached. An experienced third reviewer (ACFDM) was consulted to resolve inconsistencies.

We extracted data from the included studies using a form that included the characteristics of primary studies (such as authors, evaluated tool, location, year of publication, target population, and questionnaire respondent). The same form also captured the methodological characteristics (sample size, period between test and retest, details about the subjective assessment, description of the objective assessment methods, and the agreement between the subjective and objective methods). We considered more than one validity estimate per study when the effects of the global estimate were not reported. For studies reporting the stratified validity estimation effects, we followed a pre-specified order of priority to assess the inclusion criteria and type of barriers and facilitators. The order was as follows: first, the study population (children, adolescents), followed by a domain of sedentary behavior (e.g., screen time, sedentary time), subjective method (questionnaire, diary), objective method (e.g., accelerometer, direct observation), and type of report (e.g., parental report, self-report).

### 2.4. Data Synthesis

Concordance is the degree to which scores or rankings are identical [15]. We adopted agreement correlation coefficients between the subjective and objective methods (reference method) evaluated concomitantly with the outcome. We captured the following characteristics related to the independent variables: type of measurement, subjective method, objective method, and type of report. Although all statistically significant outcomes were reported, the heterogeneity among the definitions and measurements across the studies impaired us from pooling the findings. Figure 1 presents the flow diagram of the review, according to the PRISMA statement.

## 3. Results

### 3.1. Study Selection and Characteristics of Eligible Studies

We identified 511 records of potentially relevant studies. Of these records, seven [16,17,18,19,20,21,22] were eligible studies for our review. The included studies were published after 2005, of which four assessed test-retest reproducibility and reliability, one assessed construct validity, and six assessed test validity. Table 1 provides a description of the included studies.

### 3.2. Characteristics of Subjective Assessment Methods Recovered

Questionnaires

Self-report was the most used method to assess SB. Likert-type scales and multiple-choice answers were most used, and dichotomous answers were the least common (Appendix A).

Barriers and facilitators of SB

Questions about the barriers and facilitators of SB are associated with several factors: parental influence on their children’s behavior, individual (disposition of screens at home), and environmental factors (climate, accessibility, safety of the neighborhood in which the individuals live) [17,18,19,22].

Parental influences on their children’s behavior

Two studies [16,17] addressed questions related to parents’ encouragement and motivation for their children to be physically active with outdoor or indoor practices. Questions assessed the time available for parents to be physically active with their children, parental encouragement for children to be physically active, and adequate environment for PA engagement (whether children have a spacious backyard). These aspects can be considered as barriers for SB.

Individual-level barriers

Promoting and encouraging the use of media as a way to monitor/control children’s behavior during meals is understood to be a facilitator of SB [16,17,22]. Other strategies, such as the use of punishment, is understood to be a barrier. Engagement in watching TV (time), and the use of computers and smartphones were the media-related aspects measured by the identified questionnaires.

Perceptions of the environment/neighborhood and the weather 

The weather was mentioned as both a facilitator and a barrier by parents. Letting the child/adolescent engage in outdoor activities on hot, humid, and cold days [16,17], and days with pleasant weather was perceived to be a barrier to SB. One study addressed neighborhood safety [16,17]. Aspects related to the perceived environment included the assessment of paths’ conditions, the presence of traffic lights and crosswalks, and accessibility to public parks, green areas, or private clubs. Having a good/excellent perception of the neighborhood was a barrier to SB. Conversely, neighborhoods with high criminal levels were perceived as a facilitator for SB. 

Reliability estimates

The Cronbach’s alpha coefficient was the most reported statistical method to assess internal consistency [16,19,20,22]. Acceptable values ranged from 0.54 to 0.88. The Kappa coefficient was reported in one study [17,18], with values ranging from 0.39 to 0.97, and it was considered acceptable for the present review [23]. Three studies [17,18,21] did not report agreement methods for internal consistency. For our review, we considered an intraclass correlation coefficient (ICC) equal to or greater than 0.75 as excellent, between 0.60 and 0.74 as good, between 0.40 and 0.59 as fair, and <0.40 as poor [24]. Ridley showed high results. The study [20] presented factors with high values, and [19] with moderate to high agreement values. Additional details are available in Table 2.

Validity estimates

Pearson’s and Spearman’s coefficients were the most used statistical methods for validity estimates [16,17,18,20,21,22]. Three studies presented a weak correlation [16,17,18,20,21], one study presented a moderate correlation [21], and one study did not report a validity estimate [20]. Table 3 demonstrates the characteristics of validity findings. Validity estimates ranged from r = 0.31 to 1.00 for self-reported questionnaires, and from r = −0.1 to 0.97 for accelerometry. Table 3 describes the descriptive characteristics of validity estimates.

## 4. Discussion

We conducted a comprehensive systematic review of studies addressing the reliability and validity of questionnaires on the perceived barriers and facilitators of SB in children and adolescents. We identified that questionnaires assessing barriers and facilitators of sedentary behavior to be moderately to strongly reliable and to have moderate to weak validity [16,17,18,19,21,22]. We also identified a lack of studies in the last two decades addressing the psychometric properties of these questionnaires, and therefore the lack of adequate tools to obtain valid and reliable reports.

Several factors influence SB, including family income [16,17], neighborhood safety [16,17], personal motivation [16,17,22], and the number of household appliances [16,19,20]. This study identified the main factors associated with SB [16,17]. In our review, SB is associated with non-school days, probably because the school environment presents opportunities for the beneficial practice of physical activity [25]. Parental education level also appears to be associated with the perception of barriers and facilitators to SB. One study [26] demonstrated that parents with a higher educational level engage more in PA with their children, compared to those with lower levels [26,27]. 

Questionnaire items about parental rules for a child’s TV-watching were the most reported. One possible explanation is that parents who enforce rules, watching TV with their child and eating meals when the TV is on, were associated with increased screen time. Limiting TV time, causing children to spend more time playing outdoors, was associated with barriers [25]. In this sense, understanding SB in children and adolescents is an essential prevention strategy for public health [26,27], since SB is highly prevalent in the pediatric population and directly associated with the development of cardiometabolic diseases, which can lead to death in early adulthood [28,29,30].

We demonstrated that questionnaires are a useful tool for assessing barriers and facilitators. Identified study designs (cross-sectional, cluster-randomized trial) varied very little across studies [16,17,18,19,20,21]. Individual analyses of reliability and validity items were found to be moderate to strong.

Primary studies demonstrated a moderate to high reliability of their questionnaires, and the most studied items related to media exposure. However, parental influence is the most crucial item for SB barriers and facilitators. 

Our review identified studies with methodological heterogeneity regarding reliability; more specifically, these differences may be related to the different accelerometer models used, the accelerometer implantation protocol, and body site location. Three studies [16,20,22] used the a GT3X ActiGraph model accelerometer, which is highly reliable [31]. The other three studies [17,18,19,21] used the MTI ActiGraph model which has an acceptable level of reliability [32,33,34]. The results from the different models of accelerometers suggest that the devices evaluate the movement similarly. However, further studies to assess intra- and inter-instrument reliability are needed.

Among the strengths of this study are the search carried out in four databases, which reflects that most of the relevant literature on the topic has been investigated. Additionally, having two independent reviewers assess the studies is another strength. An aspect that can be viewed as a limitation is the fact that all the studies were performed in high-income countries. Although this is not a limitation regarding the conduction of the review, it demonstrates a lack of representation of middle to low-income countries. The heterogeneity of the included studies impeded us from conducting a pooled analysis, although this is a result of the state-of-the-art literature and not a limitation of the review process. 

This study illuminated the issues influencing SB by showing that this issue affects both developed and developing countries. Moreover, our review provided valuable information on how parental behavior is directly associated with their children’s lifestyles [35,36]. Future studies on this theme in low- and middle-income countries are warranted, considering that most of these studies were carried out in North America and Europe [36,37], therefore, impeding the generalizability of these findings to other populations. Adequate, reliable, and valid instruments to assess the perceived barriers and facilitators of SB are needed given that the exposure of SB has detrimental effects on the health of children and adolescents [37].

## 5. Conclusions

Our findings suggest that questionnaires to assess the perceived barriers and facilitators of SB presented a moderate to high reliability and weak to moderate validity. Additionally, the family environment is the main factor associated with SB among children and adolescents.

## Figures and Tables

**Figure 1 ijerph-19-16834-f001:**
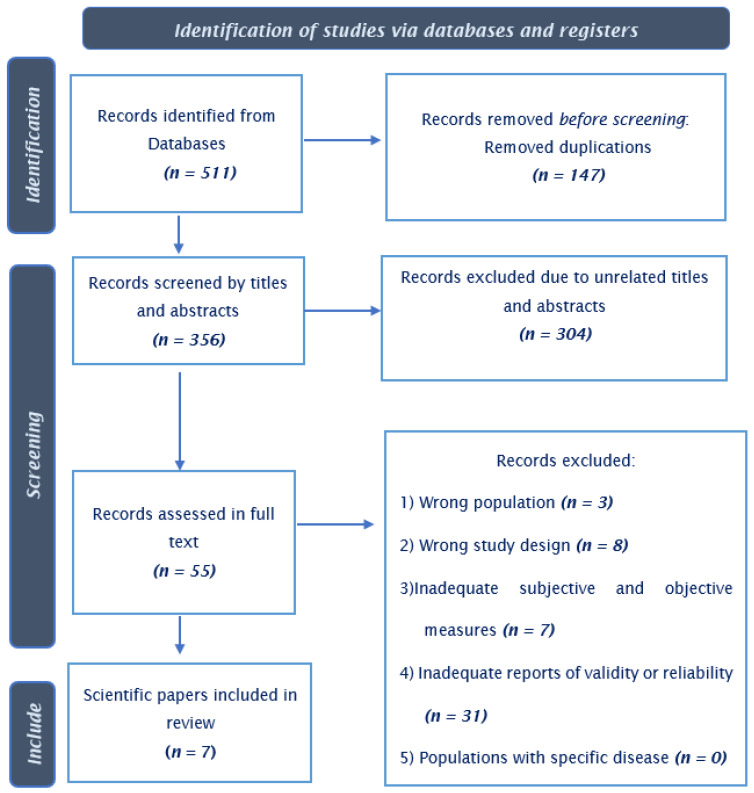
Flow diagram of systematic review process.

**Table 1 ijerph-19-16834-t001:** Description of the included studies (*n* = 7).

Study ID	Location	Journal Published	*n*	Age (Years)	Females (%)	Study Design
Janz et al., 2005 [18]	Netherlands	Res Q Exerc Sport	204	4–7	55.3	Cross-sectional
Ridley et al., 2006 [21]	Australia	Int J Behav Nutr Phys Act	1429	9–15	51.1	Cross-sectional
Jago et al., 2009 [19]	US	Int J Behav Nutr Phys Act	714	11.3	49.1	Cross-sectional
Dwyer et al., 2011 [17]	Australia	Int J Behav Nutr Phys Act	105	4–5.9	48.0	Cross-sectional
Vaughn et al., 2013 [16]	US	Med Sci Sports Exerc	324	2–5	NA	Cross-sectional
Norman et al., 2018 [20]	Stockholm County	Health Educ Behav	229	5.8–7.1	51.5	Cluster-randomized trial
Fillon et al., 2022 [22]	France	Int J Environ Res Public Health	103	8–18	52.5	Cross-sectional

**Table 2 ijerph-19-16834-t002:** Descriptive characteristics of the reliability estimates.

Study ID	Sample Size	Length of Reliability Test	Subjective Tool	Number of Questions in the Questionnaire	Internal Consistency (Test)	Test Results	Test-Retest Reliability	Reliability Findings
Janz et al., 2005 [18]	72	NR	(NAPQ) Questionnaire	1	NA	NA	(1) Coefficient kappas (κ) (2) Spearman correlation (rho)	**(1) κ = 0.39** **(2) rho = 0.30 to 0.66**
Ridley et al., 2006 [21]	32	two times in the same day	(MARCA) Questionnaire	1	NA	NA	(1) ICC	**(1) ICC = 0.88 to 0.94**
Jago et al., 2009 [19]	555	NR	(PASE) Questionnaire	24	(1) Cronbach’s alpha (α)	**(1) (α) = 0.84**	(1) Cronbach’s alpha/ Person-separation reliability	**(1) full scale 0.75–0.90**
Dwyer et al., 2011 [17]	103	two weeks apart	(Pre-PAQ) Questionnaire	7	NA	NA	(1) ICC (2) Coefficient kappas (κ)	**(1) ranged from 0.31–1.00 (ICC (2, 1))** **(2) κ = 0.60–0.97**
Vaughn et al., 2013 [16]	303	NR	NR	7	(1) Cronbach’s alpha (α)	**(1) (α) = 0.54–0.88**	NA	NA
Norman et al., 2018 [20]	229	NR	(EPAQ) Questionnaire	2	(1) Cronbach’s alpha (α)	**(1) α = 0.87**	(1) Cronbach’s alpha (α)	**Factor 1 (** **α = 0.81)** **Factors 2 (** **α = 0.79)** **Factors 3 (** **α = 0.77)**
Fillon et al., 2022 [22]	103	7 days	(CAPAS-Q) Questionnaire	31	(1) Cronbach’s alpha (α)	**(1) α = 0.71 and 0.68**	(1) Lin’s concordance correlation coefficient	(1) 0.193 ep = 0.076

Note: Not reported: NR. Not applicable: NA. Questionnaire: Preschool-age Children’s Physical Activity Questionnaire (Pre-PAQ), Eating and Physical Activity Questionnaire (EPAQ), Physical Activity Self-Efficacy Questionnaire (PASE), Netherlands Physical Activity Questionnaire (NPAQ), Multimedia Activity Recall for Children and Adolescents (MARCA), Physical and Sedentary Activity Questionnaire for Children and Adolescents (CAPAS-Q), Factor 2 * = Limit your child to watching TV, DVDs, or playing on the computer, smartphone or tablet for 2 h a day at the most? In bold are the most significant results.

**Table 3 ijerph-19-16834-t003:** Descriptive characteristics of the validity estimates and psychometric properties of the questionnaires.

Study ID	Sample Size	Length of Validity Test	Objective Assessment	Units of Measurement	Validity Estimate	Validity Findings
Janz et al., 2005 [18]	204	4 days	accelerometer	min/day	Spearman correlation	0.16
Ridley et al., 2006 [21]	66	1 day	accelerometer	min/day	Spearman correlation	**0.36 to 0.45**
Jago et al., 2009 [19]	83	5 days	accelerometer	min/day	ICC	(r = 0.17–0.33)
Dwyer et al., 2011 [17]	67	4 days	accelerometer	min/day	Pearson correlation	0.19–0.28
Vaughn et al., 2013 [16]	303	4 days	accelerometer	min/day	Pearson correlation	−0.1 to 0.08
Norman et al., 2018 [20]	NR	7 days	accelerometer	min/day	Pearson correlation	NA
Fillon et al., 2022 [22]	103	7 days	accelerometer	min/day	Pearson correlation and Spearman correlation	NR

Note: NR: Not reported. NA: Not applicable. In bold are the most significant results.

## Data Availability

The datasets supporting the conclusions of this article can be accessed by reasonable request to the Corresponding author: Augusto César F. De Moraes.

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
