# Peer review of "Validity and Reliability of Questionnaires That Assess Barriers and Facilitators of Sedentary Behavior in the Pediatric Population: A Systematic Review"

_ijerph, 2022, doi:10.3390/ijerph192416834_

Round 1
Reviewer 1 Report (Previous Reviewer 1)
Congratulations for your work
Author Response
Thank you for rating our article.
Reviewer 2 Report (New Reviewer)
ID: ijerph-2036085
Title: Validity and reliability of questionnaires that assess barriers and facilitators of sedentary behavior in the pediatric population. A systematic review.
Thank you for providing a chance to review this manuscript.
Comment: Rejection or Major Revision.
Detailed information:
Abstract
Line 31: “Four studies (66.6%) reported correlation coefficients”. I don't quite understand the meaning of this sentence. What kind of correlation? Please be more specific!
Introduction
Line 48-53: The phenomenon of sedentary behavior in pediatric population needs more background description, and it is suggested to show more data to illustrate the prevalence of this problem more intuitively.
Line 55-60: What is the connection between this research and yours? The research object of this article is the pediatric population, so please focus on the pediatric population.
Line 61-62: “Questionnaires to measure the SB of children and adolescents are widely disseminated in the literature” Since there are a lot of literatures, why only six literatures were finally selected in this paper? This is very confusing!
Methods
Search Strategy
Line 93-100: The retrieval time is not clearly described. Please clarify the start time and the end time of the literature search.
Figure 1
The proportion of irrelevant articles (n=301) is too high, I doubt whether your search strategy is effective!
Results
Barriers and facilitators of sedentary behavior
Line 169-193, Page 5: The content described in this section is consistent, and it is recommended that these results be merged into one result.
Table 2-3
From these results, the reliability and validity of the final six studies have not achieved the desired results. I once again doubt whether the final six literatures you selected are representative!
Discussion
Line 301-302: “We demonstrated that questionnaires are a useful tool to assess barriers and facilitators' characteristics and are commonly used in lifestyle research.” I don't think this paper comes to this conclusion!
Line 302-303: “The sample size, gender, and age distribution are satisfactory.” What are the criteria for sample size, gender, and age distribution satisfaction? Please explain in the methods.
Overall: This part discusses the results and relevant reasons of the previous research at a large length. And what's your suggestion? This is an indispensable part of systematic review!
Please read the PRISMA list carefully. In my opinion, the biggest problem of this article is that there are too few studies included in the final analysis, so it is difficult to prove the purpose of this paper. Then, rephrase your sentences to make your expressions clear, especially the introduction and discussion sections. Furthermore, there are many basic errors in this article, such as grammar, punctuation and so on. Last but not least, finding a native English speaker to improve the writing can considerably improve the quality.
Thank you and my best,
Your reviewer
Round 2
Reviewer 2 Report (New Reviewer)
ID: ijerph-2036085
Title: Validity and reliability of questionnaires that assess barriers and facilitators of sedentary behavior in the pediatric population. A systematic review.
Thank you for providing a chance to review this manuscript. The authors have revised most of the recommendations, and there are some errors remaining. Please review the manuscript carefully to better improve the quality of the paper.
Comment: Major Revision.
Detailed information:
Methods
Search Strategy
The retrieval time is still not clear. What I want to know is the publication time of the literature you collected.
Figure 1
Does the setting of inclusion criteria lead to a high proportion of irrelevant literature? What is the basis for inclusion criteria? Is it reasonable?
Overall: Just as you say, few articles fit your criteria. And as I have repeatedly mentioned, is the time and scope of literature search appropriate for this study? Are the inclusion criteria reasonable? This is the core of a systematic review and requires a lot of thought.
Results
Supplementary Tab.
1) There is little description about this part. What is the main point of this result? 2) It is suggested to describe the research results and significance of each item or dimension detailly.
Discussion
“The sample size, gender, and age distribution are satisfactory.” What are the criteria for sample size, gender, and age distribution satisfaction? Please explain in the methods. Otherwise, my opinion is not to delete this sentence in the manuscript simply, but it needs to be clearly described in the method. Please read my opinion carefully.
Thank you and my best,
Your reviewer
Author Response
Please see the attachment.

This manuscript is a resubmission of an earlier submission. The following is a list of the peer review reports and author responses from that submission.
Round 1
Reviewer 1 Report
Dear authors,
I congratulate you for the work as I consider it interesting and necessary, but some improvements should be made on it.
Abstract
It would be interesting to know if there were any exclusion criteria. And how many articles were found in total and how many after applying the inclusion and exclusion criteria. In line 25 they only indicate the totals after the review.
Include in the abstract that you searched for articles with both subjective and objective measures of sedentary behaviour.
Introduction
In general the introduction is too short and the need for a review is not clear. Try to express it better
Line 48 - should explain more deeply the relationship of sedentary lifestyles to health and disease in general and then to children in particular in an extensive way, there is a lot of research on this.
Line 55 - Talk more about the paediatric questionnaires on sedentary lifestyles. What was the first one, how many items they have and if they have been validated in one or several languages, compare them and give data on them, if they are defined for different age groups or not as physical activity is very different at one age or another, etc.
Methods
Data extraction
Line 110- Explain if the reviewers have experience as reviewers before.
Line 115-126 Why don't you make a table with all these questions from the 6 final articles and attach them in the results section? I recommend it.
Line 138-141- the explanation of the Flow diagram should be done in the methodology section.
Figure 1 is not cited in the text, better to put the Flow diagram in the methodology, it is not a reuslt.
Results
Table 1 is not cited in the text, neither is Table 2.
Line 149-187- indicates which are the study variables, the most used questionnaires, etc. but it would be interesting to say exactly the number of studies that indicate that such a variable is related to another variable. Indicate numerically all these appreciations of these lines.
The legend of tables 2 and 3 could be better formatted and explained.
Discussion
Line 216- I think you should try to infer why these results are the way they are. Give your opinion as authors as to why these variables are related to sedentary lifestyles and what the relationship between them might be.
Line 230- Indicate which studies have the highest reliability and what are the findings of these studies, it is not clear which variables have been most extensively studied or have the highest reliability.
Line 242- Indicate future lines of research.
Conclusion
Adequate.
Reviewer 2 Report
This review paper was the result of a comprehensive search that resulted in only 6 papers to include. The importance of the review is not clear to me as written, and the specific comments below center around how it could be made clearer.
- Introduction: Please state the objective of this review clearly at the end of the introduction to orient the reader to the paper’s purpose. For example, is the objective to examine the degree to which questionnaires for SB have been validated, or to understand factors related to SB?
- Since only 6 studies were reviewed, it may be possible to summarize in a table the degree to which the findings were consistent across studies with respect to factors influencing SB. This would make it easier for the reader to understand to what degree the factors examined overlapped between studies, and where there was overlap, if the findings from different studies agreed or disagreed with one another. In other words, did all studies that looked at factor X show that factor X was associated with SB, where X could be eating in front of the TV, for example.
- Please note the type of accelerometer that was used for each study, as there are known differences in reliability between devices. Please comment in the discussion on the reliability of the accelerometers and how that might impact the results of the studies.
- Discussion: regarding the statement “We demonstrated that questionnaires are a useful tool to assess barriers and facilita-226 tors' characteristics and are commonly used in lifestyle research,” one could argue that the researchers who published the 6 reviewed papers did this. Can you comment on relative strengths and weaknesses of the questionnaires that were evaluated?
Minor comments:
- Methods: Recommend spelling out what the E and C in PECO stand for.
- Table 1: The last row appears to be a duplicate (study by Norman et al 2018 is already listed)
- Table 2 is difficult to interpret, and this is particularly true of the last column. Consider separating the results for internal consistency and reproducibility to make the table easier to read.
- In both Tables 1 and 2, consider highlighting the strong associations in some way to aid with interpreting the results.
Round 2
Reviewer 1 Report
Dear authors,
Please mark in the article underlined in yellow the modifications you have made in the manuscript.
You must respond to the review in a blinded manner, without any author details or affiliation.
And you must respond to each change, line or suggestion one by one, as you have left many unanswered or you do it globally so that no changes appear.
They should also put a title in an affirmative sentence and not a question.
Make the changes indicated in revision 1 plus the ones in this revision and mark it so that we can do a thorough job.
Thank you very much.
Reviewer
Reviewer 2 Report
In the revision of this paper, the authors were somewhat responsive to comments, but significant concerns remain. The authors have added that "The aim of our review was to explore the validity and reliability of questionnaires that address barriers and facilitators for SB in children and adolescents aged 2 to 19 years and the main associated factors." However, Table 2 is unchanged and I still find it difficult to read and interpret. The authors did not provide clarity regarding whether there was consistency across studies in factors related to SB. There also seem to be conflicting messages in the paper, as it states both that "questionnaires are a useful tool to assess barriers and facilitators' characteristics" and that "questionnaires to assess barriers and facilitators of sedentary behavior have weak to moderate validity and reliability for measuring barriers and facilitators of sedentary behavior." The second statement suggests limited usefulness. Due to the lack of clarity and the limited responsiveness to the previous review, I recommend rejecting the paper.